# Charge Exchange Spectroscopy of Multiply Charged Erbium Ions

Yuki Nishimura [1,*], Saki Imaizumi [1], Hajime Tanuma [1], Nobuyuki Nakamura [2], Yuichiro Sekiguchi [3], Shinya Wanajo [4], Hiroyuki A. Sakaue [5], Daiji Kato [5], Izumi Murakami [5], Masaomi Tanaka [6] and Gediminas Gaigalas [7]

1 Department of Physics, Tokyo Metropolitan University, Tokyo 192-0397, Japan
2 Institute for Laser Science, The University of Electro-Communications, Tokyo 182-8585, Japan
3 Department of Physics, Toho University, Chiba 274-8510, Japan
4 Max-Planck-Institut für Gravitationsphysik (Albert-Einstein-Institut), 14476 Potsdam, Germany
5 National Institute for Fusion Science, Gifu 509-5292, Japan
6 Astronomical Institute, Tohoku University, Sendai 980-8578, Japan
7 Institute of Theoretical Physics and Astronomy, Vilnius University, 10257 Vilnius, Lithuania
* Correspondence: nishimura-yuki2@ed.tmu.ac.jp

**Abstract:** The origin of heavier elements than iron is still under discussion, and recent studies suggest that the contribution of the r-process in neutron star mergers is dominant. Future modeling of such processes will require a huge amount of spectroscopic data on multiply charged ions of heavy elements. However, these experimental data are extremely scarce for heavy elements. In this work, we have performed the measurements of charge exchange spectroscopy for multiply charged Er ions in the visible light range. We report observed emission lines from multiply charged Er ions and their identification based on theoretical estimates.

**Keywords:** charge exchange; transitions in the visible light range; erbium ions; kilonova

## 1. Introduction

The "r-process" during the binary neutron star merger is regarded as a most promising candidate for the mechanism of synthesis of heavier elements than iron in the universe [1]. For the first time, the binary neutron star merger was detected by gravitational-wave detectors Ligo and Virgo on 17 August 2017, and the corresponding transient called a kilonova was observed by optical telescopes [2]. To understand the observed spectrum from the kilonova via a radiative transport simulation, spectroscopic data of heavy elements are necessary. In the previous work, we predicted that triply and lower charged ions of heavy elements contributed to the photo-emission from kilonovae [3]. In particular, lanthanoid elements significantly contribute to visible and infrared emissions. However, spectroscopic data for heavy element atoms are still limited, and we must use roughly estimated atomic data for the radiative transfer simulations with heavy elements.

In this work, we have performed the measurements of charge exchange spectroscopy to provide atomic data for erbium ions, one of the lanthanoid elements that significantly contribute to visible and infrared emissions from kilonovae. This method is very useful for observing emission lines of charge-selected ions. We report new lines from charge exchange spectroscopy in collisions of multiply charged Er ions with neutral gases, and show the results of line identification of $Er^{3+}$ ions using the results of a theoretical calculation with the Cowan code [4].

## 2. Experiment

The multiply charged Er ions were produced in a 14.25 GHz electron cyclotron resonance ion source (ECRIS) by introducing an Er rod into the oxygen plasma at Tokyo

Metropolitan University [5]. The ion beam was extracted from the plasma by an electric potential difference of 15 kV, and directed into the collision cell after the charge-state separation with a dipole magnet. The beam intensity was measured with a Faraday cup located behind the collision region. Typical mean ion currents during the experiments were 10–50 nA. The cell was filled with a thin target gas with a pressure that satisfied the single collision conditions. The background pressure and gas pressure in the collision cell were about $10^{-7}$ and $10^{-3}$ Pa, respectively. Emissions after charge exchange reactions due to the ion–gas collisions were guided to the entrance of the bundled optical fiber with a length of 10 m through a convex lens with a diameter of 40 mm and a focal length of 81.3 mm, and were transferred to a Czerny–Turner-type spectrometer (iHR 320, HORIBA). The bundled optical fiber cable consists of 26 fibers each of 250 μm diameter and the transmission loss is 0.6 dB/m at 260 nm. The focal length of the spectrometer is 320 mm. A liquid-nitrogen-cooled CCD camera (Spec-10:400B/LN, Princeton Instruments) was installed with the spectrometer to observe the photon emission in the visible light region. The slit width of the spectrometer was about 200 μm, and a grating with 1800 gr/mm and a blaze wavelength of around 500 nm was used for measurements. The full width at half maximum was about 0.4 nm in the spectra, and the wavelength determination accuracy was about ±0.25 nm after the calibration with a mercury lamp. In this experiment, the wavelength range of more than 30 nm in the first-order diffraction was passed by the monochromator and recorded by the CCD camera simultaneously. We indicate the wavelength at the center of the observed range as $\lambda_c$ in this manuscript. The exposure time was 10–60 minutes, and the spectra recorded on the CCD were integrated after removing cosmic ray noises.

We used $Er^{4+}$ and $Er^{5+}$ as the projectile ions and Ar and $N_2$ as the target gases. These projectile ions consist of natural $^{166}Er$. Those two targets have almost the same value for the ionization potential energy, namely 15.76 eV for Ar and 15.6 eV for $N_2$. The dominant electron capture levels of the projectile ions should be determined by the ionization potential of the target according to the simple understanding of the charge exchange reaction between highly charged ions and neutral gas targets. Therefore, photon emissions from the projectiles should be similar in collisions with Ar and $N_2$ targets. If the observed emission spectra show strong target dependence, it will be due to the emission from the targets. Using these simple principles, we can easily distinguish the emission lines of Er ions from those of targets. Since both Ar and $N_2$ have many electrons, we must consider not only the emission lines due to the single electron capture but also those due to double electron capture by comparing the spectra measured for different charge states of projectiles.

## 3. Results

In Figure 1, we show the observed emission spectra in the collisions of $Er^{4+}$ with Ar and $N_2$ in different wavelength regions. The common emission lines with two different targets can be regarded as the photon emissions from Er ions. On the other hand, the photon emissions from the targets might be observed in the particular target case.

The lines at wavelengths of 316.7 and 347.6 nm in Figure 1a, 423.1, 429.2, and 454.4 nm in Figure 1b, 482.2, 484.1, and 490.5 nm in Figure 1c, and 590.3 nm in Figure 1d were observed in collisions with two targets and should be regarded as the emissions from Er ions.

Figure 2 shows the observed spectra in the collisions of $Er^{5+}$ with Ar and $N_2$ in different wavelength regions. The lines at 423.1, 438.1, and 454.4 nm in Figure 2a, 490.5 nm in Figure 2b, 545.0, 546.2, and 550.7 nm in Figure 2c, and 584.7 and 589.9 nm in Figure 2d should be the emissions from Er ions.

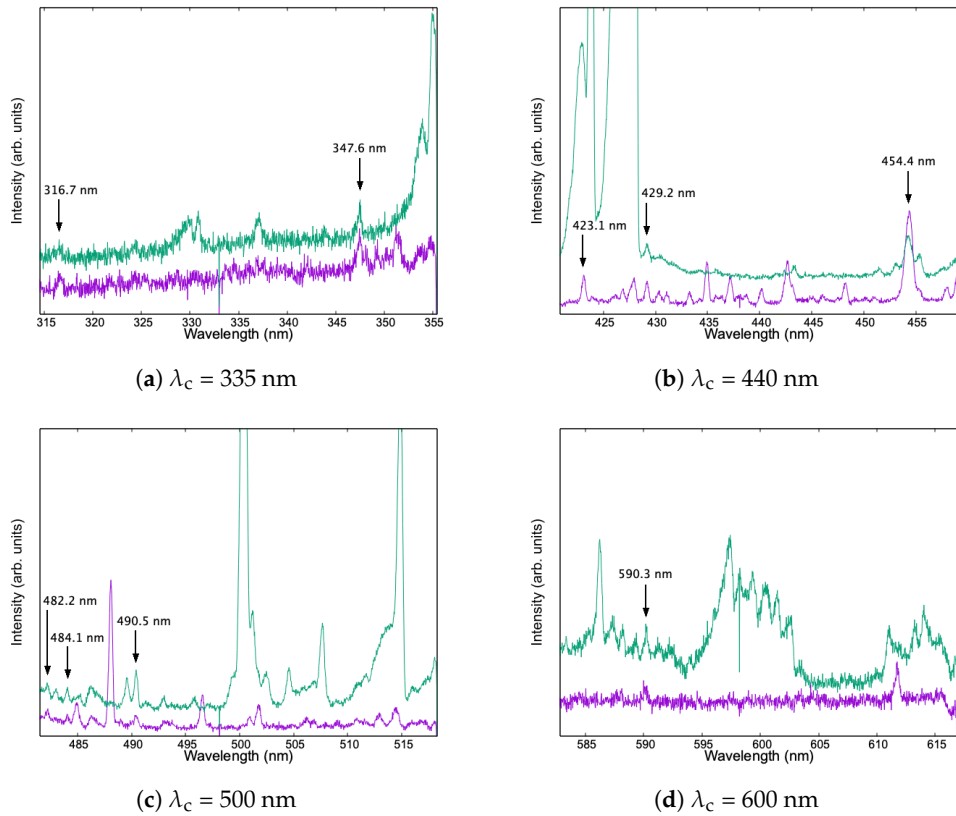

**Figure 1.** Visible emission spectra of radiation resulting from the collisions of $Er^{4+}$ ions with Ar (purple) and $N_2$ (green) targets at an energy of 60 keV.

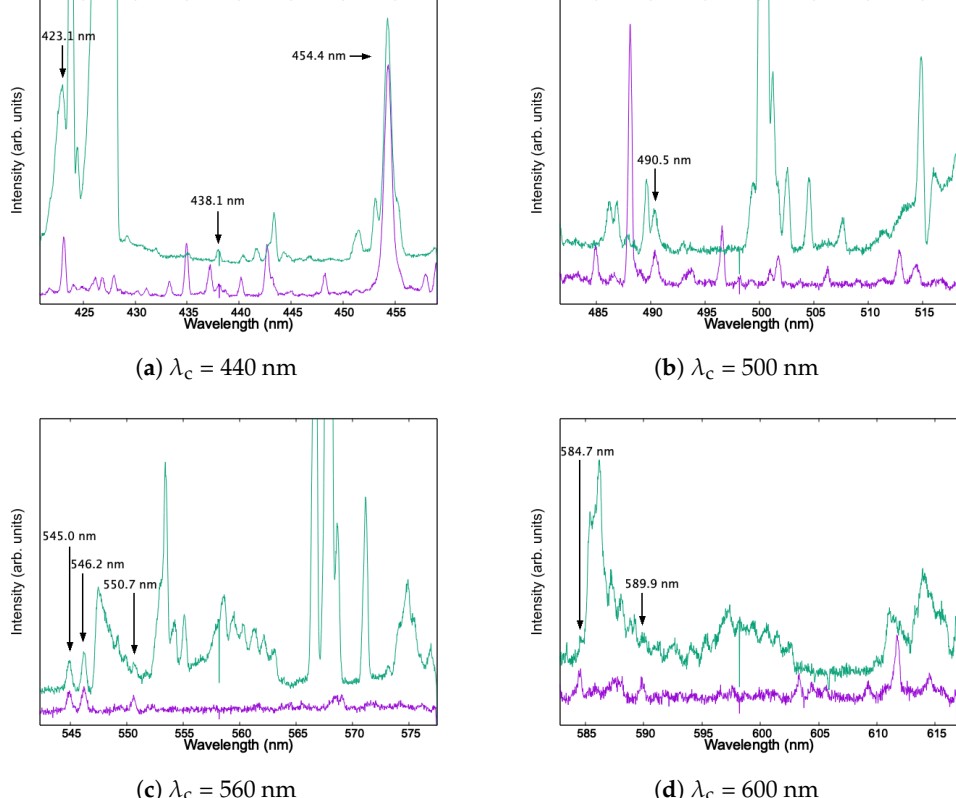

**Figure 2.** Visible emission spectra of radiation resulting from the collisions of $Er^{5+}$ ions with Ar (purple) and $N_2$ (green) targets at an energy of 75 keV.

In every panel in Figures 1 and 2, we find many sharp lines in Ar target spectra and many emission bands in $N_2$ target ones. We conclude that these lines correspond to the emissions from $Ar^+$ according to comparison with the NIST Atomic Spectra Database (ASD) [6]. The broad and strong emissions produced in the $N_2$ target are due to the $N_2^+$ first negative system [7]. Emissions from ionized targets that are produced in collisions of highly charged ions with neutral gas targets are often observed and known as the transfer excitation (TE) process. The TE process involves two electrons and is more complicated than the simple one-electron transfer reaction.

We did not find any data on highly charged Er ions tabulated in the NIST ASD, but some information on Er I–III is available in this database. Comparing the transition wavelengths of $Er^{2+}$ in the database with those measured, we consider that the line emissions at 316.7, 429.2, and 590.3 nm are due to $Er^{2+}$ produced in the double electron capture to $Er^{4+}$. Furthermore, comparing the spectra with $Er^{4+}$ and $Er^{5+}$ injections, we find that the lines at 423.1, 454.4, and 490.4 nm are commonly observed and these might correspond to the $Er^{3+}$, which is produced by the single electron capture in $Er^{4+}$ collisions and by the double electron capture in $Er^{5+}$ ones.

Finally, we identified that the line emissions at 347.6, 423.1, 454.4, 482.2, 484.1, and 490.5 nm are due to $Er^{3+}$ and that the 438.1, 545.0, 546.2, 550.7, 584.7, and 589.9 nm lines are emitted from $Er^{4+}$.

## 4. Discussion

The energy levels of $Er^{3+}$ were calculated by means of the Cowan code (Kramida's PC version) and confirmed by comparison to vacuum spark spectra in the wavelength range of 70.5–246 nm [4]. Using these energy levels, we have tried to search the upper and lower energy levels corresponding to our observed transition wavelengths of $Er^{3+}$, taking the selection rule of E1 transitions into consideration.

Table 1 shows the results of the assignments of the transitions to the experimental wavelengths. The uncertainty of this calculation is estimated to be ±6 nm in the visible region, which is deduced from the finding that the difference between the experimental and calculated level energies lie in the range between −137 and 111 $cm^{-1}$ in the reference paper. Several transitions become candidates for the lines at 347.6, 423.1, and 490.5 nm. On the other hand, the same transition is assigned as the candidate for the lines at 482.2, 484.2, and 490.5 nm. Unfortunately, we can not find any transition corresponding to the 454.4 nm line.

Figure 3 shows the energy levels of $Er^{3+}$. Because of the electronic interactions, each configuration has a huge number of fine-structure levels and large energy spreads. In this figure, we show the possible upper levels of transitions in Table 1 as red lines.

**Table 1.** Observed transition wavelengths of $Er^{3+}$ and possible upper and lower levels of the transitions. $\lambda_{\text{exp}}$ means the vacuum wavelengths measured in the experiments. $J$, $E$, $\Delta E$, and $\lambda_{\text{cal}}$ mean the total angular momentum, the energy value, the difference between the energy of the upper level and the lower levels, and the calculated wavelength using the energy difference $\Delta E$, respectively.

| $\lambda_{\text{exp}}$ (nm) | Upper Level | $J_{\text{upper}}$ | $E_{\text{upper}}$ (cm$^{-1}$) | Lower Level | $J_{\text{lower}}$ | $E_{\text{lower}}$ (cm$^{-1}$) | $\Delta E$ (cm$^{-1}$) | $\lambda_{\text{cal}}$ (nm) |
|---|---|---|---|---|---|---|---|---|
| 347.7 | 4f$^{10}$6p ($^5$I)$^6$H | 7.5 | 147,062.3 | 4f$^{10}$5d d($^3$M)$^4$L | 8.5 | 117,818.7 | 29,243.6 | 341.955 |
|  | 4f$^{10}$6p ($^5$I)$^6$H | 7.5 | 147,062.3 | 4f$^{10}$5d d($^3$L)$^4$L | 7.5 | 118,103.2 | 28,959.1 | 345.315 |
|  | 4f$^{10}$6p ($^5$I)$^6$H | 7.5 | 147,062.3 | 4f$^{10}$5d d($^3$M)$^4$K | 7.5 | 118,356.0 | 28,706.3 | 348.356 |
|  | 4f$^{10}$6p ($^5$I)$^4$K | 8.5 | 147,473.8 | 4f$^{10}$5d d($^3$M)$^4$K | 7.5 | 118,356.0 | 29,117.8 | 343.433 |
| 423.2 | 4f$^{10}$5d d($^5$I)$^4$I | 5.5 | 94,497.7 | 4f$^{11}$ $^2$G2 | 4.5 | 70,623.3 | 23,874.4 | 418.859 |
|  | 4f$^{10}$5d d($^5$G)$^4$D | 1.5 | 116,918.7 | 4f$^{11}$ $^2$F1 | 2.5 | 93,368.7 | 23,550.0 | 424.628 |
|  | 4f$^{10}$5d d($^5$G)$^4$H | 3.5 | 117,197.3 | 4f$^{11}$ $^2$F1 | 2.5 | 93,368.7 | 23,828.6 | 419.664 |
| 454.5 | n/a |  |  |  |  |  |  |  |
| 482.4 | 4f$^{10}$5d d($^5$I)$^6$L | 5.5 | 91,148.2 | 4f$^{11}$ $^2$G2 | 4.5 | 70,623.3 | 20,524.9 | 487.213 |
| 484.2 | 4f$^{10}$5d d($^5$I)$^6$L | 5.5 | 91,148.2 | 4f$^{11}$ $^2$G2 | 4.5 | 70,623.3 | 20,524.9 | 487.213 |
| 490.6 | 4f$^{10}$5d d($^5$I)$^6$L | 5.5 | 91,148.2 | 4f$^{11}$ $^2$G2 | 4.5 | 70,623.3 | 20,524.9 | 487.213 |
|  | 4f$^{10}$5d d($^3$P)$^4$F2 | 2.5 | 113,690.4 | 4f$^{11}$ $^2$F1 | 2.5 | 93,368.7 | 20,321.7 | 492.085 |

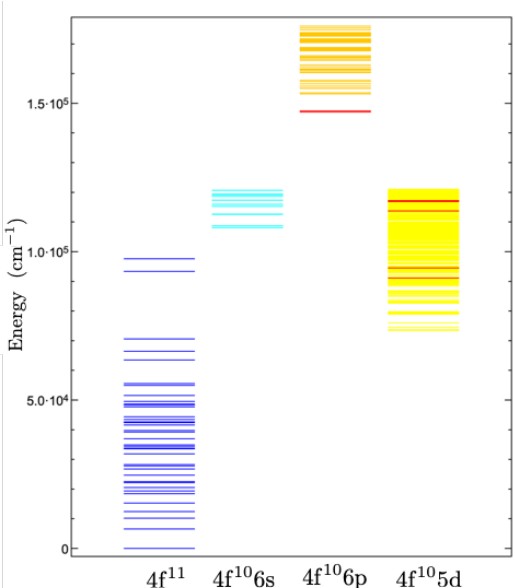

**Figure 3.** Energy levels of $Er^{3+}$ calculated with the Cowan codes [4]. The candidates of the upper levels for the observed transitions are shown as red lines in this figure. The energy levels at 147,062.3 and 147,473.8 cm$^{-1}$ and those at 116,918.7 and 117,197.3 cm$^{-1}$ cannot be shown as separated lines because of their small energy differences.

In Table 1, the minimum and maximum energies of the upper levels are $E_{min} = 91,148.2$ cm$^{-1} = 11.30$ eV and $E_{max} = 147,473.8$ cm$^{-1} = 18.28$ eV, respectively. Here, we apply the simple potential crossing model for the charge exchange reaction. If we consider only the Coulombic repulsion between two positive ions in the single electron transfer process, the potential curve crossing distance $R_x$ is given by

$$R_X = \frac{1}{4\pi\varepsilon_0}\frac{(q-1)e^2}{Q} \tag{1}$$

where $q$ is the incident ion charge state and $Q$ is the energy gain, which is given by the following relationship:

$$Q = IP_p - IP_t + EX_p$$

where $IP_p$ is the ionization potential of the projectile ion with the charge state of $q - 1$, $IP_t$ is the ionization potential of the neutral target, and $EX_p$ is the excitation energy level of the projectile ion produced in the single electron capture process. The ionization energies of $Er^{3+}$ and the target are 42.4 and 15.6 eV, and $EX_p$ is expected in the range of 11.30–18.28 eV, as mentioned above. Therefore, Q will be between 8.5 and 15.5 eV, and we can estimate the crossing distance of 2.8–5.1 Å in this collision system, which agrees with the reaction window in charge transfer reactions of highly charged ions [8]. This result shows that the electron capture levels do not contradict the understanding of the ordinary theory for charge exchange reactions in collisions of multiply charged ions with neutral gaseous targets. However, the electronic states of $Er^{3+}$ are very complicated and a huge number of optical transitions between the fine-structure levels should be expected. Therefore, our finding that only a few transitions are observed is difficult to understand without accurate quantum mechanical calculation for the collision dynamics.

**Funding:** This work was in part supported by JSPS KAKENHI from the Japan Society for the Promotion of Science, Grant Number JP19H00694. One of the authors (YN) is grateful to the Japan Science and Technology Agency for support in the establishment of university fellowships towards the creation of science technology innovation, Grant Number JPMJFS2139.

**Conflicts of Interest:** The authors declare no conflict of interest.

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
