# Peer review of "Charge Exchange Spectroscopy of Multiply Charged Erbium Ions"

_atoms, doi:10.3390/atoms11020040_

Round 1

Reviewer 1 Report

The authors report in “Charge exchange spectroscopy of multiply charged erbium ions” the identification of multiple spectral lines in Er^3+. Such spectral lines are key ingredients to understand astrophysical processes and their potential sites, which are believed to be the origin of heavy elements in the universe. On the other hand, spectral lines for highly charged ions are scarcely found in data bases, in particular those exhibiting open shells due to the complexity of theory and experiment. 

In their approach, the authors produced multiply charged Er ions using an ECR source, selected them with a magnet, and subsequently investigated their light emission that was initiated by gas collisions. To this end they used a CCD camera attached to a spectrograph. 

The analysis of the spectra is convincing and the theory approach to identify the transitions seems to be suitable. A firm assignment of these transitions to certain level pairs is rather difficult in many cases due to the complexity of the electronic structure. 

The manuscript is well written, and the setup is explained in detail. The title is fine, and the conclusions are backed up by experiment and theory. In my view, this work merits publication. In the following, I will give some final suggestions for the authors to be considered before acceptance.

Major improvements: 

I was a little surprised to see no form of schematic overview of the setup. I would encourage the authors to consider adding such a figure, which could for example include the sources, the magnet, the optics, relevant distances etc.

I suggest the authors to add information on the focal length of the spectrometer, the length of the bundled optical fiber and on its transmission efficiency in the relevant wavelength range.

Please indicate in which diffraction order the spectra shown in Fig.1/2 were measured?

Please indicate the pressures used in the collision cell.

Please indicate that nat. Er has been taken for the experiments if this is the case.

The authors expect when the ionization potentials of the gaseous species (Ar, N2) are similar the electron capture processes in a highly charged projectile would be similar in a simplified picture of charge exchange process. The authors then conclude the photon emissions to be similar as well. I advise the authors to be more precautious with the latter statement as even if the same orbital was occupied during an EC process the interaction with Ar and N2 will be different leading eventually to different quenching channels and thus enabling cascades that were previously suppressed.

Which version of the Cowan codes have been used?

I suggest the authors to give more precise characteristics for line identification instead of using “target/projectile dependence”. One would be interested in knowing the boundary conditions in terms of SNR threshold to include/exclude certain peaks.

Not all upper levels can be identified in Fig.3. Could the authors improve the figure and assign the levels?

Could the authors indicate how they estimated the energy gain used in Eq.1?

Minor corrections:

Line 4: “star merger called” -> “star merger event called” 

Line 8/9 -> One may reformulate it like: “We report observed emission lines … and their identification based on theoretical estimates.”

Line 15: “object” -> “transient”

Line 27: delete “gasses” -> “gases”

Line 42: “26 fibers each of 250”

Line 53: “according to a simple” 

Line 61: “spectra measured for different”

Line 64+71: “N2 in different”

Line 75: delete “can”

Line 76: “can see” -> “conclude”

Line 78: please reformulate the sentence!

Line 79: “targets in” -> “targets that are produced in”

- Line 83: “can not see” -> “did not found”;  “from NIST” -> “tabulated in the NIST”

- Line 84: “available from” -> “available in”

- Line 85: “, we can consider” -> “with those measured, we consider”

Line 91: “. and ” -> “ and that ”

- Tab. 1: Please define J, Delta E, lambdas! Are the given wavelengths air wavelengths?

Reviewer 2 Report

Reviewer’s Comments:

The manuscript “Charge exchange spectroscopy of multiply charged erbium ions” is very interesting work. This paper investigates the origin of heavier elements than iron has been discussed up to now, and recent studies suggest that the contribution of the r-process in neutron star mergers is dominant. Comparing theoretical radiation transport calculations with spectra from the optical counterpart associated with a neutron star merger called “kilonova”, we can verify the synthesis of heavy elements. Highly reliable radiative transfer simulation requires a huge amount of spectroscopic data on multiply charged ions of heavy elements. However, these experimental data are extremely scarce for heavy elements. However, the following issues should be carefully treated before publication.

1. In abstract, the author should add more scientific findings.

2. Keywords: the synthesized system is missing in the keywords. So, modify the keywords.

3. In the introduction part, the introduction part is not well organized and cited references should cite recently published articles such as 10.3389/fchem.2022.1023316,  10.3390/molecules27196457

4. Introduction part is not impressive and systematic. In the introduction part, the authors should elaborate the scientific issues in the multiply charged erbium ions research.

5. Results…, The author should provide reason about this statement “In Figure 1, we show the observed emission spectra in the collisions of Er4+ with Ar and N2 at different wavelength regions”.

6. The authors should explain regarding the recent literature why “Table 1 shows the results of the assignments of the transitions to the experimental wavelengths”.

7. Discussion. The author should explain the latest literature “Therefore, we can estimate the crossing distance of 2.8–5.1 Å in this collision system, which agrees with the reaction window in charge transfer reactions of highly charged ions”.

8. The author should provide reason about this statement, “Therefore, our finding that only a few transitions are observed is difficult to understand without accurate quantum mechanical calculation for the collision dynamics”.

9. Comparison of the present results with other similar findings in the literature should be discussed in more detail. This is necessary in order to place this work together with other work in the field and to give more credibility to the present results.

10. The conclusion part is very week. Improve by adding the results of your studies.
